# Trends in pediatric cochlear implants: The dual impact of COVID-19 and Lebanon's crisis

**Joy M. El Maalouf[1], Maged T. Ghoche[1], Gabriel Dunya[2]\***

**1** Department of Otolaryngology, Gilbert and Rose-Marie Chagoury School of Medicine, Lebanese American University, Beirut, Lebanon, **2** Department of Otolaryngology, American University of Beirut Medical Center, Beirut, Lebanon

\* gd15@aub.edu.lb

## Abstract

### Background

Hearing loss is a major public health issue globally, especially in low- and middle-income countries (LMICs), where access to cochlear implants (CIs) is restricted by cost and limited healthcare infrastructure. In Lebanon, the 2020 economic crisis and the COVID-19 pandemic further reduced access to essential services, including pediatric CIs.

### Aim

This study assesses the impact of the 2020 Lebanese economic crisis on the number and funding sources of pediatric CI surgeries.

### Methods

A retrospective review was conducted on 228 pediatric patients who underwent 235 CI surgeries between 2017 and 2023. The number of surgeries and funding sources were compared before and after the 2020 crisis. Funding categories included government, private insurance, donations, or a combination. Data were analyzed using R software.

### Results

There was no significant difference in the number of surgeries before (113) and after (122) the crisis (p = 0.56). However, a marked shift in funding occurred. Government-funded procedures dropped from 45.87% to 12.61% (p < 0.001), while private and donation-based funding rose from 32.11% to 66.39% (p < 0.001). The mean age at surgery declined from 5.86 to 3.57 years post-crisis (p < 0.05), indicating greater awareness of early intervention benefits.

**Data availability statement:** All relevant data are within the paper and its Supporting Information files.

**Funding:** The author(s) received no specific funding for this work.

**Competing interests:** The authors have declared that no competing interests exist.

## Conclusion

Despite economic hardship, the demand for pediatric CIs persisted, with families turning to private and charitable sources. Enhanced government support and the classification of CIs as essential health services are vital. Early diagnosis and intervention should be prioritized to improve outcomes in children with hearing loss.

## Introduction

Hearing loss is the fourth leading cause of disability worldwide, drastically disturbing individuals' quality of life and their ability to communicate efficiently [1]. Untreated hearing loss in early childhood can lead to permanent damage such as developmental delays, social isolation, and increased unemployment rates, resulting in substantial economic impacts on both individuals and society as a whole [2–5]. The burden of hearing loss is mostly marked in low- and middle-income countries (LMIC), where access to essential hearing assistance, including cochlear implants, is often limited due to various systemic barriers [6].

The higher incidence of deafness in LMIC can be attributed to a range of non-genetic factors, such as maternal infections during pregnancy, prematurity, and low birth weight [7]. These factors can lead to significant hearing impairment, which, if not addressed early, can have lifelong consequences [8]. While cochlear implants have revolutionized treatment options for many individuals with hearing loss [8], allowing them to access sound and improve their communication abilities, their utilization remains below 15% globally [9].

The average cost of cochlear implants, along with the associated surgical procedures, in LMIC typically ranges from $15,000 to $40,000 [10]. This figure does not account for ongoing expenses related to maintenance, replacement parts, batteries, and necessary postoperative care. Such high costs place cochlear implants out of reach for many families, exacerbating existing inequalities and limiting opportunities for children with hearing impairment [11].

Furthermore, governments in developing countries frequently neglect the provision of hearing impairment services, often focusing resources on more immediate health crises or life-saving interventions [12]. Securing funding for cochlear implants is challenging, as many stakeholders prioritize expenditures that address acute health needs over long-term developmental interventions. Despite evidence supporting the cost-effectiveness of cochlear implants in improving quality of life, the financial burden remains largely on the shoulders of families, many of whom are already facing economic hardships [13,14].

This was also seen during the COVID-19 pandemic, where elective surgeries, including CI procedures, were deprioritized due to strain on healthcare systems. As resources were redirected to urgent care, many CIs were postponed, affecting patients who rely on these procedures for hearing and quality of life [15].

The aim of this study is to evaluate the impact of the 2020 Lebanese economic crisis and the COVID-19 pandemic on the number and financial coverage of pediatric

cochlear implant (CI) surgeries in Lebanon. Specifically, we investigate changes in the frequency of CI procedures and shifts in funding sources, such as government support, private insurance, and donations, before and after the crisis. By doing so, we aim to highlight the financial pressures placed on families and underscore the importance of sustainable health policy responses to ensure equitable access to this essential intervention in Lebanon and similar LMIC. This paper also aims to raise awareness about these challenges and advocate for stronger, more consistent support from governmental bodies to safeguard access to critical healthcare services like CI surgery.

## Methodology

### Study design

This is a retrospective quantitative study that aims to analyze the impact of the Lebanese 2020 economic crisis on the funding and access to cochlear implants (CIs) in the pediatric Lebanese population at 18 years of age or younger. This paper compared the type of financial coverage and the number of CI procedures done pre- and post-Lebanese economic crisis, shedding light on the main limitations in access to this essential medical procedure.

We asked three different Lebanese cochlear implant companies, representing the different cochlear implant manufacturers (Advanced Bionics, Cochlear and MED-EL), to provide us with the following data on pediatric patients who underwent the latter procedure: demographics (governorate, gender, and the age at which CI was performed), the healthcare facility where the surgery was performed (university versus non-university versus non-Lebanese university hospital), the year in which CI was done (data grouped into two periods: 2017–2019 (before 2020) and 2020–2023 (after 2020)), and the financial coverage type (government-funded, private insurance, donations (charities, NGOs, etc.), or combinations of more than one).

For the analysis, governmental coverage, insurance coverage, and the combined "governmental and private" coverage were grouped under the category Third Party Coverage. Private coverage, donation-based coverage, and the combined "private and donation" coverage were grouped under the category Private Coverage.

The collected data was statistically analyzed using "R statistics" to detect changes in the number of surgeries performed before and after the onset of the economic crisis. The analysis emphasized on comparing the frequency and distribution of CI across different governorates, hospitals, genders, and age groups. Moreover, we examined any statistically significant shifts in the source of financial coverage used before and after the crisis. This helped establish whether the economic crisis has posed a limitation to accessing cochlear implants.

All data were anonymized in order to protect the privacy of the patients' data. Ethical approval was provided from the American University of Beirut Medical Center (AUBMC)'s Institutional Review Board (IRB) - BIO-2024–0252, and data collection was in compliance with applicable data protection regulations. The data were obtained from cochlear implant (CI) companies, and the research team had no direct contact with patients or their guardians. The need for informed consent was waived by the IRB due to the retrospective nature of the study and the use of fully anonymized secondary data. The data were accessed for research purposes on September 15, 2024.

This retrospective quantitative study aimed to analyze the impact of Covid-19 and the 2020 Lebanese economic crisis on the funding and access to cochlear implants (CIs) for the pediatric Lebanese population aged 18 years or younger. This paper compared the type of financial coverage and the number of CI procedures performed before and after the Lebanese economic crisis, highlighting the main limitations in accessing this essential medical procedure.

## Results

A total of 228 patients were included in this study, who collectively underwent 235 surgeries between 2017 and 2023. Bilateral cochlear implants were performed on 7 of these patients. The average age of the participants was $4.67 \pm 3.41$ old and patients' ages ranged from 0 till 15 years of age.

All surgeries were performed at various institutions across Lebanon, including both university and non-university hospitals. We divided these patients into two separate groups: those who underwent CI before 2020 and those in 2020 onward (Table 1).

## Trends in cochlear implant surgery rates

The analysis of cochlear implant (CI) surgeries from 2017 to 2023, as depicted in Fig 1, reveals a fluctuating trend in the number of procedures over the study period. From 2017 to 2019, the number of CI surgeries showed a steady increase. However, starting 2019 until 2021, a significant decline in the number of surgeries was observed, to reach less than half the number of CI in 2019. After 2021, a noticeable upward trend in the number of CI surgeries was recorded, with the number of surgeries gradually rising each year until 2023. This upward trajectory suggests a gradual recovery in cochlear implant procedures, following the sharp decline observed in 2019.

To further investigate the impact of the observed decline in CI surgeries during the study period, the average number of surgeries before 2020 and 2020 onward was compared. The average number of surgeries before the crisis (2017–2019) was 30.7, while that from 2020 onward was 30.8 (Fig 2). A Z-test for the comparison of means was conducted to assess whether this difference was statistically significant. The test yielded a p-value of 0.119, indicating that there was no statistically significant difference in the average number of surgeries between the two periods.

To gain a deeper understanding of the factors influencing the trends in CI surgeries, the CI's coverage was analyzed to explore the effect of both the COVID-19 pandemic and the ongoing economic crisis on the type of coverage.

The bar graph shows a significant shift in cochlear implant surgery coverage from government and insurance to private sources and donations after 2020. The odds ratio of receiving CI were about five times higher for those covered by private sources or donations in the post-2020 period (OR = 5.13, 95% CI: [2.66, 10.26]), with the difference being statistically significant ($p < 0.05$) (Fig 3).

**Table 1. Overall characteristics of the study population.**

| | Description | Before 2020 | After 2020 | Overall | p |
|---|---|---|---|---|---|
| **Number of surgeries** | | 113 | 122 | 235 | 0.56 |
| **Number of patients** | | 109 (4 bilateral) | 119 (3 bilateral) | 228 | 0.51 |
| **Mean Age (SD)** | | 5.86 | 3.57 | 4.67 (3.41) | <0.001 |
| **Gender (%)** | Male | 58 (53.21) | 57(47.9) | 115 (50.44) | 0.54 |
| | Female | 51 (46.79) | 62 (52.1) | 113 (49.56) | 0.54 |
| **Governorate (%)** | Beirut | 34 (30.09) | 7 (5.74) | 41 (17.45) | 0.0028 |
| | South | 29 (26.66) | 27 (22.13) | 56 (23.83) | |
| | North | 25(22.12) | 28 (23.95) | 53(22.55) | |
| | Bekaa | 14 (12.39) | 11 (9.02) | 25 (10.64) | |
| | Mount Lebanon | 8 (7.08) | 15 (12.30) | 23 (9.79) | |
| | International | 2 (1.77) | 8 (6.65) | 10 (4.26) | |
| **Hospital Type (%)** | University | 71 (65.14) | 59 (49.58) | 130 (57.02) | <0.0001 |
| | Non-University | 38 (34.86) | 62 (52.1) | 10 (43.86) | |
| | NA | 2 (1.83) | 1(0.84) | 3(1.32) | |
| **Hospital Location (%)** | Beirut | 49(44.95) | 89(74.79) | 138(60.53) | 0.48 |
| | Non-Beirut | 16(14.68) | 21(17.65) | 37(16.23) | |
| | NA | 46(42.2) | 12(10.08) | 58(25.44) | |
| **Coverage (%)** | Third party | 0 (0.0) | 3 (2.52) | 3 (1.32) | 0.03 |
| | Private | 2 (1.83) | 3 (2.52) | 5 (2.19) | |
| | NA | 3 (2.75) | 3 (2.52) | 6 (2.63) | |

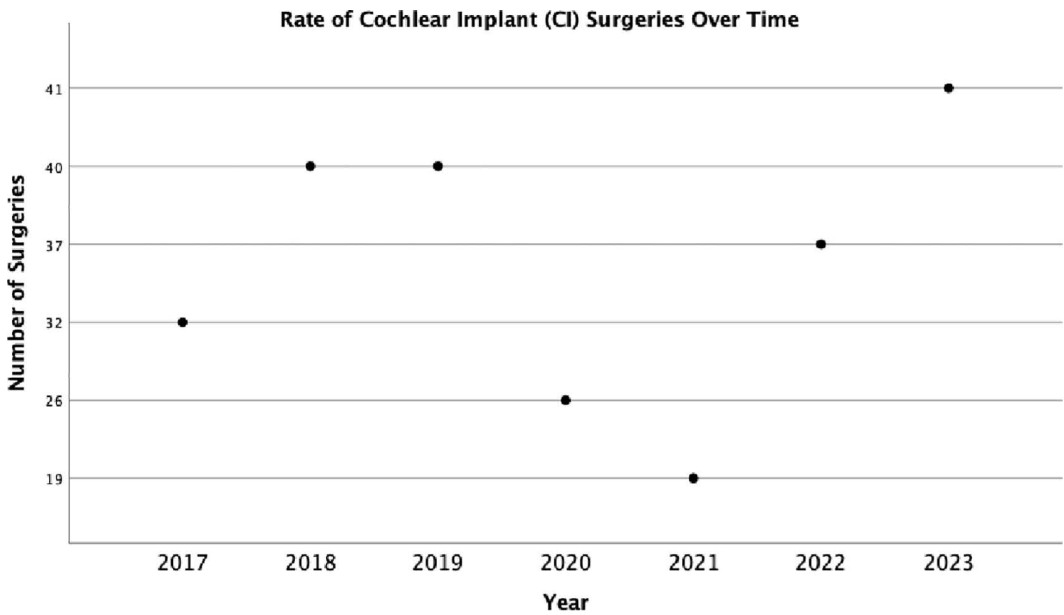

**Fig 1. Graph showing the change in the rate of cochlear implant surgeries from over time.**

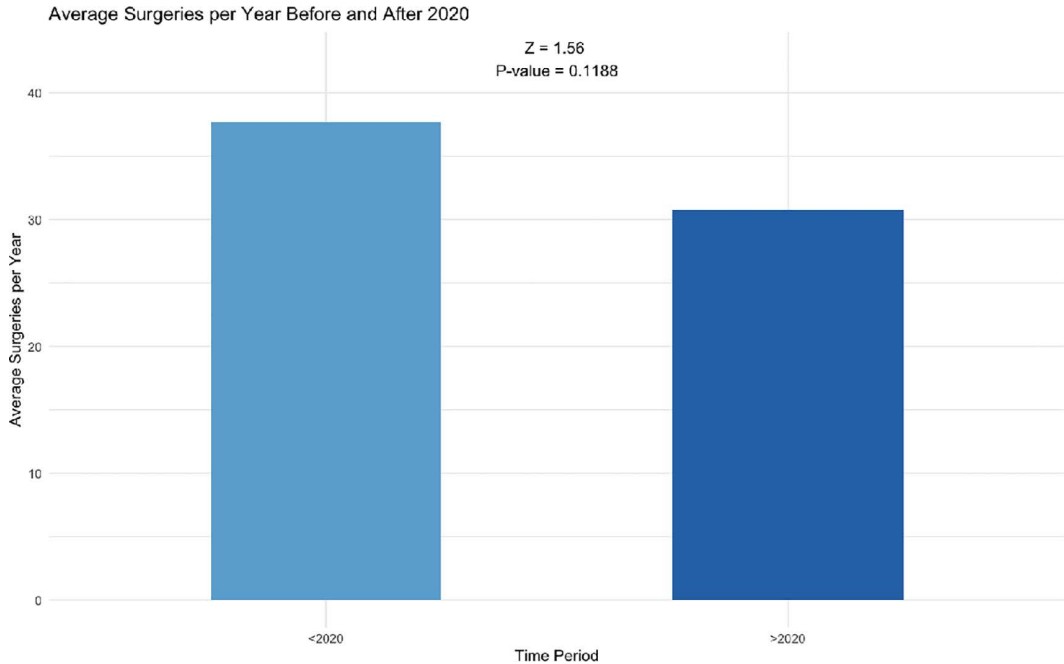

**Fig 2. Bar chart comparing the average number of cochlear implant surgeries: Pre-2020 vs. 2020 and Beyond.**

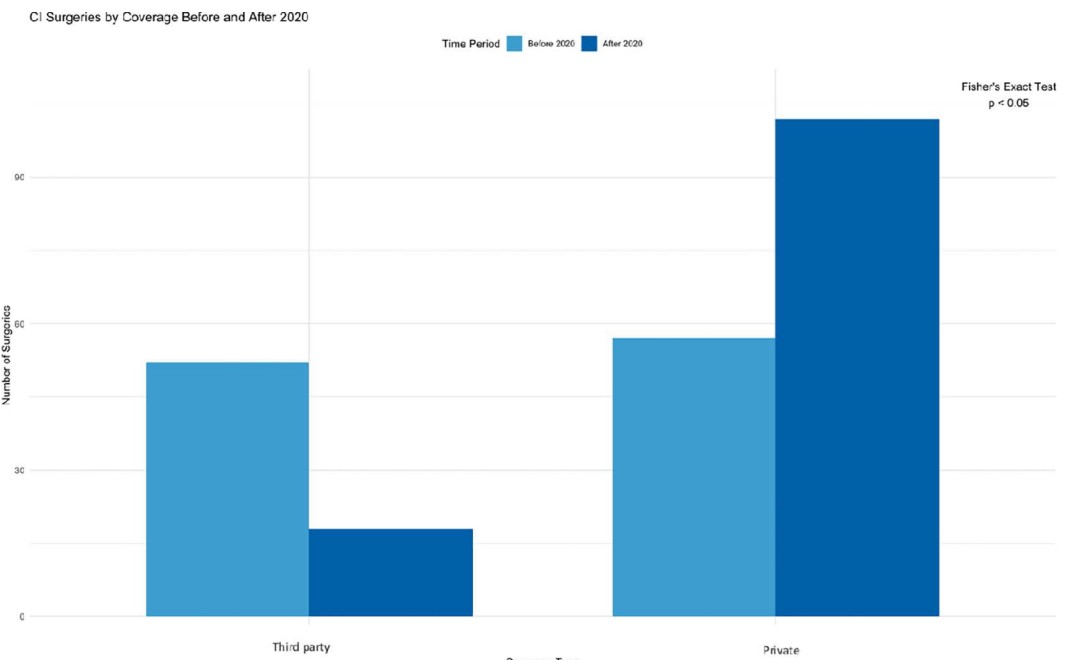

**Fig 3. Bar chart comparing the rate of cochlear implant surgeries by coverage: Pre-2020 vs. 2020 and Beyond.**

Building on the previous analysis of coverage trends, we examined how the type of hospital—university-affiliated versus non-university—affects the coverage for cochlear implant surgeries. In non-university hospitals, only 16% covered by government or insurance. In university-affiliated hospitals, private coverage and donations were also the primary sources, reflecting a similar trend (Fig 4).

Further analysis of coverage type based on patient location revealed that, in general, both Beirut (Capitol) and non-Beirut patients were predominantly covered by private insurance or donations. However, government and insurance coverage were more common among patients residing in Beirut. It is worth noting that many patients were unable to provide their location of residence, which may affect the overall distribution of coverage types.

A Chi-squared test revealed no significant difference in coverage types between patients in Beirut and those outside of it, with a p-value of 0.09794. This suggests that the distribution of coverage for cochlear implant surgeries is similar regardless of patient location (Fig 5).

The examination of age trends in cochlear implant surgeries revealed a significant decrease in the mean age, from 5.86 years before 2020 to 3.57 years after 2020, with a p-value $< 0.05$. This suggests a shift toward younger patients receiving cochlear implant surgeries after 2020 (Fig 6).

The analysis examined changes in the age at which cochlear implant surgeries were performed before and after 2020. The results revealed a significant decrease in the mean age, from 5.86 years before 2020 to 3.57 years after 2020, with a p-value $<0.05$ (Fig 6).

## Discussion

Cochlear implants have revolutionized the management of sensorineural hearing loss, particularly in the pediatric population, offering significant improvements in speech and language development. However, the high cost of the procedure remains a major challenge, limiting accessibility for many families.

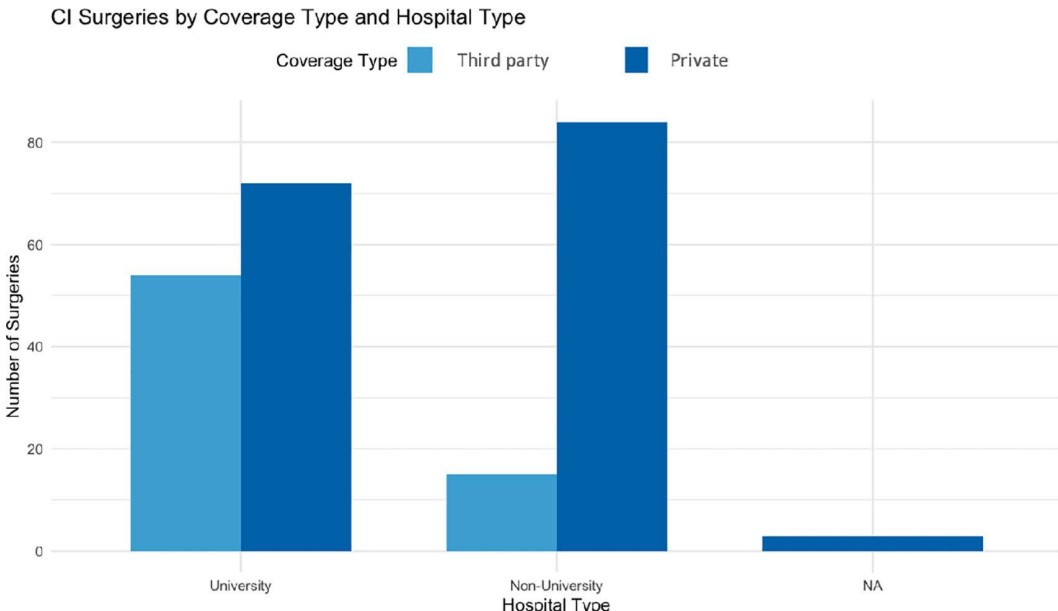

**Fig 4. Bar chart showing the association between hospital type (University/Non-university/NA) and coverage.**

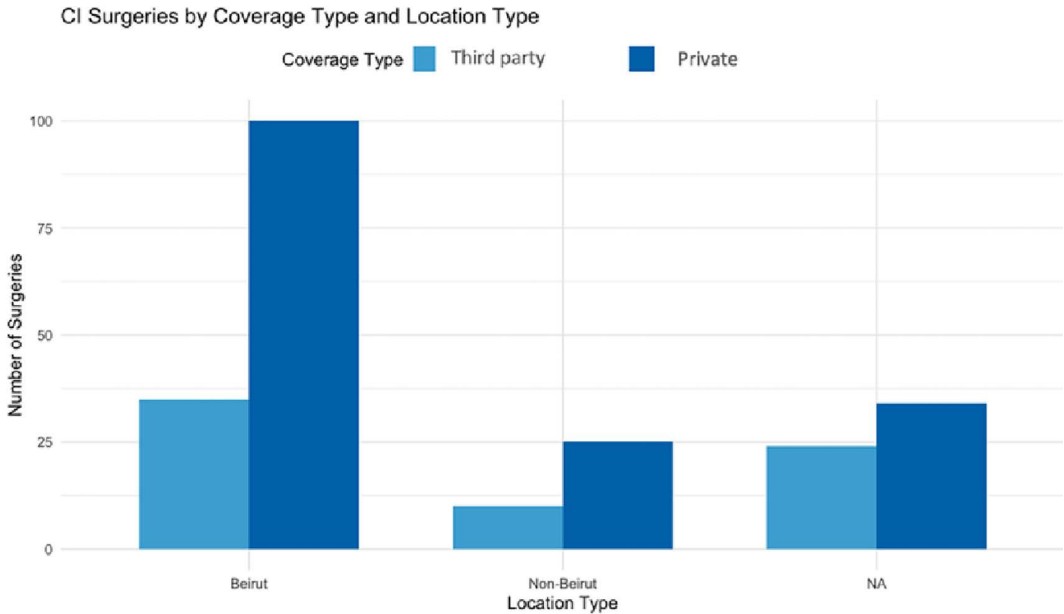

**Fig 5. Bar chart showing the association between hospital location (Beirut/Non-Beirut) and coverage.**

In fact, the average price of cochlear implant device and surgery in LMIC ranges from $15,000 to $40,000 [10]. Nevertheless, the cost of the implant does not only depend on the device itself but its maintenance, replacement, and batteries, aside from the preoperative and postoperative care. Thus, treatment can cost more than $40 000 [11].

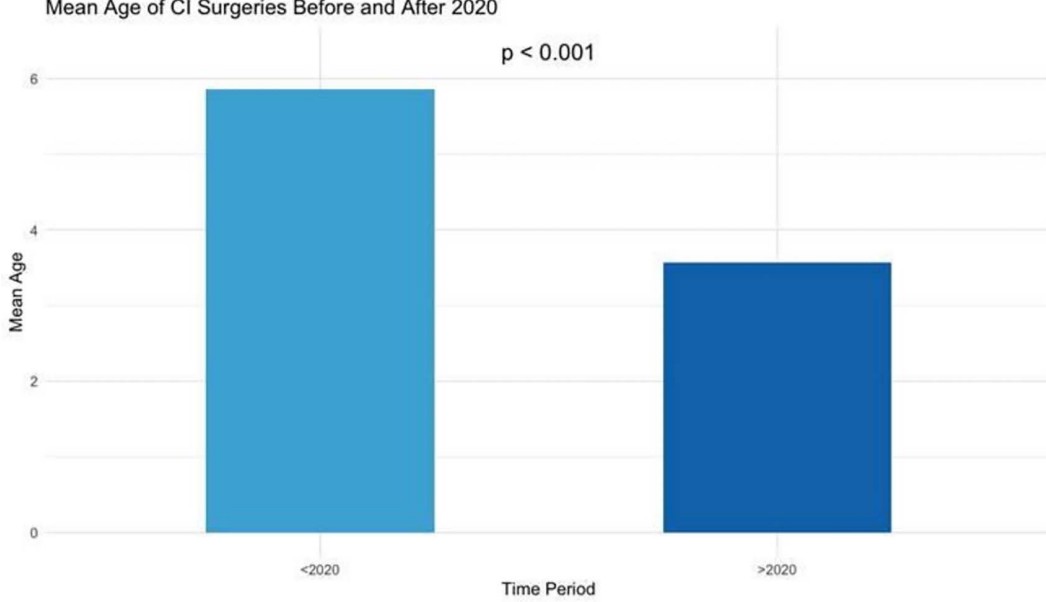

**Fig 6. Bar chart comparing the mean age of patients: Pre-2020 vs. 2020 and Beyond.**

Governments in developing countries commonly neglect hearing impairment [12,16]. Hence, although many LMIC such as India, China, Saudi Arabia, Nigeria, Pakistan, Sub-Saharan Africa, and Latin America have performed CIs, they have been faced with numerous challenges [17–19].

These findings are surprisingly consistent with trends observed in developed countries, where similar challenges in accessing cochlear implants exist despite more robust healthcare systems. This may be attributed to the fact that governments in both developed and developing countries often prioritize resources for more immediate, life-saving health interventions over long-term developmental services such as hearing impairment treatments [12,16]. Securing funding for cochlear implants is particularly difficult, as many healthcare systems prioritize expenditures that address acute health crises. Despite the substantial evidence supporting the cost-effectiveness of cochlear implants in improving patients' quality of life, the financial burden remains predominantly on families.

Tarabichi et al have noticed that in contrast to the developing countries, European governments have been playing a major role in funding CI for the last 50 years. The World Health Organization attributes the lack of governmental involvement in the financial health sector to political and ministerial inefficiency in low-income countries [20].

In fact, LMIC have less public funding, and those available are minimally spent on the health segment. Likewise, private insurances have minor role in financially supporting health conditions and surgeries in low-income countries compared to middle or high-income countries [21]. Consequently, CI burden still falls on the citizens of developing countries.

Furthermore, the COVID-19 pandemic led to significant disruptions in healthcare, resulting in the postponement of elective surgeries to conserve resources and minimize virus exposure [22]. On March 15, 2020, the Veterans Administration (VA) implemented a nationwide pause on elective surgeries, following the issuance of triage guidelines by the American College of Surgeons the next day [23].

The COVID-19 pandemic led to a noticeable decrease in cochlear implant surgeries in 2020, primarily due to disruptions in healthcare services. While the number of surgeries declined in some regions, other areas showed a slight increase. This shift underscores the broader impact of the pandemic on elective procedures like cochlear implants [15].

This literature is consistent with our findings in Lebanon, where similar challenges related to the high cost and limited accessibility of cochlear implants persist.

Before the onset of the economic crisis and COVID-19, Lebanon's healthcare system was already fragile, with limited resources for covering the cost of surgeries and medical care for its citizens. The healthcare system relied heavily on a combination of private insurance, donations, and out-of-pocket payments, leaving many people without adequate financial protection [24].

When the crisis hit, with many losing their jobs, access to bank savings being restricted, and the devaluation of the Lebanese pound, patients were left unable to cover their medical expenses. At the same time, the Lebanese government, already strained by financial challenges, was unable to step in to fill the gap, further deepening the financial crisis within the healthcare sector and placing additional strain on both patients and hospitals [24].

The results of this study underscore the significant impact of Lebanon's ongoing economic crisis and the COVID-19 pandemic on cochlear implant (CI) surgeries. Between 2019 and 2021, a sharp decline in surgeries was observed, reflecting the strain on Lebanon's healthcare system. However, the steady increase in surgeries after 2021 highlights the resilience and adaptability of the Lebanese population in navigating these challenges. Despite the crisis, patients have found ways to continue vital procedures, demonstrating a capacity to recover and adjust under difficult circumstances.

One of the most remarkable outcomes is the persistent dependency on private coverage and donations for funding CI surgeries, both before and after the crisis. This reflects a longstanding topic not only in Lebanon's healthcare system but other developing countries, where the government has not prioritized CI surgeries as fundamental procedures. The economic downturn only exacerbated this gap, leaving private funding as the primary means of access.

The decline in cochlear implant surgeries in Lebanon during the 2020 crisis was primarily driven by the challenges posed by COVID-19, which led to widespread disruptions in healthcare access. However, the severe economic downturn, characterized by hyperinflation and currency devaluation, significantly worsened the situation. While the pandemic itself was the main factor contributing to the decline, the economic crisis exacerbated the issue by limiting families' financial resources, making it harder to afford the procedure. This was reflected in the shift from government to private insurance coverage, as many families turned to private options, which often did not fully cover the costs of the surgery.

Private coverage was the primary source of funding for cochlear implant surgeries across both university and non-university hospitals, with non-university hospitals more reliant on private insurance. Patients in Beirut, where major hospitals are concentrated, were more likely to have government or insurance coverage, while those in rural areas primarily relied on private insurance. Overall, the trend highlights the Lebanese healthcare system's heavy dependence on private funding, particularly for specialized procedures like cochlear implants, regardless of geographic location.

This is not only noticed when it comes to the coverage of CI, but also in other medical fields. As reported by Sanayeh et al., during Lebanon's economic crisis, hospitals have faced significant financial challenges due to reduced government funding, making it increasingly difficult to cover the costs of surgeries and leading to greater reliance on private sources and donations for healthcare coverage [25].

Additionally, an important observation that emerged from our analysis, and one that warrants attention, is the notable shift in the mean age of patients undergoing cochlear implant (CI) surgeries. This shift likely reflects an increasing awareness among Lebanese parents of the importance of early intervention for pediatric hearing loss. Cochlear implants are most effective when implanted at a younger age, and this trend suggests that, despite the economic challenges, parents are prioritizing early treatment for their children to improve long-term outcomes. The growing awareness of the benefits of early implantation may also indicate a broader shift in attitudes toward healthcare, with more focus on early intervention to enhance quality of life. In fact, Tomblin et al. (year) found that early cochlear implantation reduces the developmental gap between children with hearing loss and their normal-hearing peers, highlighting that earlier implantation leads to better language outcomes [26].

## Limitations and future directions

One important limitation of our study is the absence of detailed socioeconomic data, such as household size, income level, and home ownership status. These variables would have provided valuable insight into the broader financial and social challenges influencing access to CI surgery. However, due to the retrospective design and the fact that data were obtained from multiple hospitals and cochlear implant companies, consistent collection of such information was not feasible. Furthermore, all data were fully anonymized in accordance with IRB requirements, and no direct patient contact was permitted to supplement the dataset. Despite these constraints, the significant shift in funding from government and insurance-based coverage to private and donation-based sources, likely reflects the growing financial burden borne by families amid the concurrent public health and economic crises. Future prospective or mixed-methods studies incorporating patient-level socioeconomic data are warranted to better capture the multidimensional barriers to equitable access and to inform targeted policy interventions.

## Conclusion

This study highlights the resilience of families and the shifting healthcare landscape in Lebanon amidst concurrent economic collapse and the COVID-19 pandemic. Despite significant financial and systemic pressures, the number of pediatric cochlear implant surgeries remained stable post-2020. However, there was a clear and statistically significant shift in funding sources, from predominantly government-based support to reliance on private insurance and charitable donations. This trend underscores a growing burden on families and highlights the urgent need for renewed governmental support and reclassification of CIs as essential healthcare services. Policymakers should prioritize sustainable funding mechanisms to ensure equitable access to hearing rehabilitation services for all children, regardless of economic conditions

## Supporting information

**S1 Data.  CI data cleaned 27_9_24.**
(XLSX)

## Author contributions

**Formal analysis:** Joy M El Maalouf, Maged T. Ghoche.

**Software:** Maged T. Ghoche.

**Supervision:** Gabriel Dunya.

**Validation:** Joy M El Maalouf, Gabriel Dunya.

**Visualization:** Joy M El Maalouf, Gabriel Dunya.

**Writing – original draft:** Joy M El Maalouf, Maged T. Ghoche.

**Writing – review & editing:** Joy M El Maalouf, Maged T. Ghoche.

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
