## [Decision Letter · Decision Letter 0]

9 Jul 2025

PONE-D-25-26378Trends in Pediatric Cochlear Implants: The Dual Impact of COVID-19 and Lebanon’s CrisisPLOS ONE

Dear Dr. Dunya,

Thank you for submitting your manuscript on the evolving landscape of cochlear implant (CI) surgery funding in Lebanon in the context of the COVID-19 pandemic and economic crisis. After a thorough review by three independent reviewers, we have reached a decision that a major revision is required before further consideration for publication.

Please address each reviewer’s comments in a point-by-point response and submit a revised version of your manuscript. We look forward to receiving your updated submission. Please submit your revised manuscript by Aug 23 2025 11:59PM. If you will need more time than this to complete your revisions, please reply to this message or contact the journal office at plosone@plos.org . Please include the following items when submitting your revised manuscript:

We look forward to receiving your revised manuscript.

Kind regards,

Toru Miwa

Academic Editor

PLOS ONE

Journal Requirements:

Reviewers' comments:

Reviewer's Responses to Questions

**Comments to the Author**

1. Is the manuscript technically sound, and do the data support the conclusions?

Reviewer #1: Partly

Reviewer #2: Yes

Reviewer #3: Yes

2. Has the statistical analysis been performed appropriately and rigorously? 

Reviewer #1: Yes

Reviewer #2: Yes

Reviewer #3: Yes

3. Have the authors made all data underlying the findings in their manuscript fully available?

Reviewer #1: Yes

Reviewer #2: Yes

Reviewer #3: Yes

4. Is the manuscript presented in an intelligible fashion and written in standard English?

Reviewer #1: Yes

Reviewer #2: Yes

Reviewer #3: Yes

5. Review Comments to the Author

Reviewer #1: The manuscript presents a timely and relevant discussion on the shift in funding sources for cochlear implant surgery in Lebanon, highlighting the transition from government to private funding following the COVID-19 pandemic and the ongoing economic crisis. While the research objective is clearly stated, the analysis would benefit from a more comprehensive exploration of key socioeconomic background factors, such as household size, annual income, and home ownership status. These elements are critical to fully understanding the broader implications of funding changes on patient access and healthcare equity. Given the current scope and limitations in contextual analysis, I regret to conclude that this manuscript may not be suitable for publication in this journal.

Reviewer #2: Conclusion is not as per the objective of the study as it highlights only the difference in CI surgery pre and post COVID pandemic. You have mentioned about importance of CI surgery in conclusion which is not relevant to your study.

Reviewer #3: The manuscript is technically sound, and the data presented adequately support the authors’ conclusions.

The authors have applied suitable statistical methods, and the analyses are clearly described and justified. The results are interpreted correctly, and the statistical approach supports the overall conclusions of the study.

The data are accessible in a clear and organized format, allowing for verification of results and further analysis if needed.76 typically ranges from $15,000 to $40,000 [10]. This Fig does not account for ongoing expenses.

The methodology is explained with enough detail to ensure reproducibility, and the results are communicated clearly, supported by suitable statistical analyses. The conclusions align well with the presented data and are generally well-supported by the findings. While some figures and tables could be clearer, they still help convey the main points effectively. The discussion places the results within the context of existing research, adding relevance and depth to the study.

6. PLOS authors have the option to publish the peer review history of their article (what does this mean? ). If published, this will include your full peer review and any attached files.

**Do you want your identity to be public for this peer review?** For information about this choice, including consent withdrawal, please see our Privacy Policy .

Reviewer #1: No

Reviewer #2: **Yes: ** Nabin Lageju

Reviewer #3: No

---

## [Author Response · Author response to Decision Letter 1]

8 Aug 2025

Dear Reviewers,

Thank you for your time and thoughtful consideration of our manuscript. We sincerely appreciate the opportunity to clarify and improve areas that required further refinement. You will find significant revisions have been made based on the received suggestions.

Below, we outline our point-by-point responses. Changes made to the manuscript are indicated in blue font, with italics denoting modified or newly added text.

Reviewers' comments:

Reviewer #1

The manuscript presents a timely and relevant discussion on the shift in funding sources for cochlear implant surgery in Lebanon, highlighting the transition from government to private funding following the COVID-19 pandemic and the ongoing economic crisis. While the research objective is clearly stated, the analysis would benefit from a more comprehensive exploration of key socioeconomic background factors, such as household size, annual income, and home ownership status. These elements are critical to fully understanding the broader implications of funding changes on patient access and healthcare equity. Given the current scope and limitations in contextual analysis, I regret to conclude that this manuscript may not be suitable for publication in this journal.

We sincerely thank the reviewer for the thoughtful and constructive feedback. We appreciate your recognition of the relevance and timeliness of our manuscript in addressing the shift in cochlear implant funding in Lebanon following the economic crisis and COVID-19 pandemic.

We fully agree that variables such as household size, annual income, and home ownership would provide valuable socioeconomic context and enrich the analysis of healthcare access and equity. However, due to the retrospective design and the fact that data were acquired from multiple hospitals and cochlear implant companies, consistent collection of such information was not feasible. Additionally, our IRB approval required full de-identification of all patient data, and no direct patient contact was permitted to retrieve supplementary information.

While these limitations restricted our ability to include detailed socioeconomic indicators, we believe the shift from government-funded to private and donation-based coverage serves as a meaningful proxy for the broader financial hardship experienced by families. The reduction in age at implantation also suggests increased parental prioritization of early intervention despite worsening conditions.

In response to this feedback, we revised the study’s aim (lines 95–101) to more accurately reflect our focus: evaluating the impact of the Lebanese economic crisis and COVID-19 on the number and financial coverage of pediatric cochlear implant (CI) surgeries. By highlighting shifts in funding sources, government, private insurance, and donations, we aim to underscore the growing financial burden on families and the need for sustainable healthcare policy in Lebanon and similar LMICs.

We have also added a paragraph to the Discussion section (lines 333–346) to explicitly acknowledge this limitation and recommend that future prospective or mixed-methods studies incorporate detailed socioeconomic variables to better characterize barriers to care.

“Aim

The aim of this study is to evaluate the impact of the 2020 Lebanese economic crisis and the COVID-19 pandemic on the number and financial coverage of pediatric cochlear implant (CI) surgeries in Lebanon. Specifically, we investigate changes in the frequency of CI procedures and shifts in funding sources, such as government support, private insurance, and donations, before and after the crisis. By doing so, we aim to highlight the financial pressures placed on families and underscore the importance of sustainable health policy responses to ensure equitable access to this essential intervention in Lebanon and similar LMIC. This paper also aims to raise awareness about these challenges and advocate for stronger, more consistent support from governmental bodies to safeguard access to critical healthcare services like CI surgery.”

“Limitations and Future Directions

One important limitation of our study is the absence of detailed socioeconomic data, such as household size, income level, and home ownership status. These variables would have provided valuable insight into the broader financial and social challenges influencing access to CI surgery. However, due to the retrospective design and the fact that data were obtained from multiple hospitals and cochlear implant companies, consistent collection of such information was not feasible. Furthermore, all data were fully anonymized in accordance with IRB requirements, and no direct patient contact was permitted to supplement the dataset. Despite these constraints, the significant shift in funding from government and insurance-based coverage to private and donation-based sources, likely reflects the growing financial burden borne by families amid the concurrent public health and economic crises. Future prospective or mixed-methods studies incorporating patient-level socioeconomic data are warranted to better capture the multidimensional barriers to equitable access and to inform targeted policy interventions”.

Reviewer #2

Conclusion is not as per the objective of the study as it highlights only the difference in CI surgery pre and post COVID pandemic. You have mentioned about importance of CI surgery in conclusion which is not relevant to your study.

We thank the reviewer for this valuable comment. We agree that the Conclusion should remain closely aligned with the primary objective of the study, which was to evaluate the impact of the Lebanese economic crisis and COVID-19 pandemic on the number and funding sources of pediatric cochlear implant surgeries.

Our original intention was to highlight the real-world implications of the observed trends, particularly the shift from government to private/donation-based funding and the resilience of families in maintaining access to care despite financial hardship. However, we understand that our previous conclusion extended beyond the scope of the presented data by focusing broadly on the importance of CI surgery.

In response, we have revised the Conclusion to focus more directly on our study findings and their implications for health policy and funding priorities in Lebanon and similar contexts.

“Conclusion

This study highlights the resilience of families and the shifting healthcare landscape in Lebanon amidst concurrent economic collapse and the COVID-19 pandemic. Despite significant financial and systemic pressures, the number of pediatric cochlear implant surgeries remained stable post-2020. However, there was a clear and statistically significant shift in funding sources, from predominantly government-based support to reliance on private insurance and charitable donations. This trend underscores a growing burden on families and highlights the urgent need for renewed governmental support and reclassification of CIs as essential healthcare services. Policymakers should prioritize sustainable funding mechanisms to ensure equitable access to hearing rehabilitation services for all children, regardless of economic conditions”.

Reviewer #3:

The manuscript is technically sound, and the data presented adequately support the authors’ conclusions. The authors have applied suitable statistical methods, and the analyses are clearly described and justified. The results are interpreted correctly, and the statistical approach supports the overall conclusions of the study. The data are accessible in a clear and organized format, allowing for verification of results and further analysis if needed. The methodology is explained with enough detail to ensure reproducibility, and the results are communicated clearly, supported by suitable statistical analyses. The conclusions align well with the presented data and are generally well-supported by the findings. While some figures and tables could be clearer, they still help convey the main points effectively. The discussion places the results within the context of existing research, adding relevance and depth to the study.

We thank the reviewer for their thoughtful and encouraging feedback. We are pleased that the methodology, statistical analyses, and interpretation of results were found to be appropriate and clearly presented. We appreciate the reviewer’s recognition of the study's clarity, reproducibility, and relevance to existing literature.

---

## [Editor Report · Decision Letter 1]

13 Aug 2025

Trends in Pediatric Cochlear Implants: The Dual Impact of COVID-19 and Lebanon’s Crisis

PONE-D-25-26378R1

Dear Dr. Dunya,

We’re pleased to inform you that your manuscript has been judged scientifically suitable for publication and will be formally accepted for publication once it meets all outstanding technical requirements.

Kind regards,

Toru Miwa

Academic Editor

PLOS ONE

Additional Editor Comments (optional):

All concerns are appropriately corrected.
---

## [Editor Report · Acceptance letter]

PONE-D-25-26378R1

PLOS ONE

Dear Dr. Dunya,

I'm pleased to inform you that your manuscript has been deemed suitable for publication in PLOS ONE. Congratulations! Your manuscript is now being handed over to our production team.

Kind regards,

on behalf of

Dr. Toru Miwa

Academic Editor

PLOS ONE